# ROCK Inhibitor (Y-27632) Abolishes the Negative Impacts of miR-155 in the Endometrium-Derived Extracellular Vesicles and Supports Embryo Attachment

**DOI:** 10.3390/cells11193178

**Published:** 2022-10-10

**Authors:** Islam M. Saadeldin, Bereket Molla Tanga, Seonggyu Bang, Chaerim Seo, Okjae Koo, Sung Ho Yun, Seung Il Kim, Sanghoon Lee, Jongki Cho

**Affiliations:** 1Laboratory of Theriogenology, College of Veterinary Medicine, Chungnam National University, Daejeon 34134, Korea; 2Research Institute of Veterinary Medicine, Chungnam National University, Daejeon 34134, Korea; 3Toolgen Inc., Seoul 08501, Korea; 4Korea Basic Science Institute (KBSI), Ochang 28119, Korea

**Keywords:** miR-155, CRISPR/Cas9, extracellular vesicles, embryo, ROCK inhibitor

## Abstract

Extracellular vesicles (EVs) are nanosized vesicles that act as snapshots of cellular components and mediate cellular communications, but they may contain cargo contents with undesired effects. We developed a model to improve the effects of endometrium-derived EVs (Endo-EVs) on the porcine embryo attachment in feeder-free culture conditions. Endo-EVs cargo contents were analyzed using conventional and real-time PCR for micro-RNAs, messenger RNAs, and proteomics. Porcine embryos were generated by parthenogenetic electric activation in feeder-free culture conditions supplemented with or without Endo-EVs. The cellular uptake of Endo-EVs was confirmed using the lipophilic dye PKH26. Endo-EVs cargo contained miR-100, miR-132, and miR-155, together with the mRNAs of porcine endogenous retrovirus (PERV) and β-catenin. Targeting PERV with CRISPR/Cas9 resulted in reduced expression of PERV mRNA transcripts and increased miR-155 in the Endo-EVs, and supplementing these in embryos reduced embryo attachment. Supplementing the medium containing Endo-EVs with miR-155 inhibitor significantly improved the embryo attachment with a few outgrowths, while supplementing with Rho-kinase inhibitor (RI, Y-27632) dramatically improved both embryo attachment and outgrowths. Moreover, the expression of miR-100, miR-132, and the mRNA transcripts of BCL2, zinc finger E-box-binding homeobox 1, β-catenin, interferon-γ, protein tyrosine phosphatase non-receptor type 1, PERV, and cyclin-dependent kinase 2 were all increased in embryos supplemented with Endo-EVs + RI compared to those in the control group. Endo-EVs + RI reduced apoptosis and increased the expression of OCT4 and CDX2 and the cell number of embryonic outgrowths. We examined the individual and combined effects of RI compared to those of the miR-155 mimic and found that RI can alleviate the negative effects of the miR-155 mimic on embryo attachment and outgrowths. EVs can improve embryo attachment and the unwanted effects of the de trop cargo contents (miR-155) can be alleviated through anti-apoptotic molecules such as the ROCK inhibitor.

## 1. Introduction

The pig is considered a crucial model for transgenic animals and xenotransplantation; however, the process of embryo production in vitro is quite challenging due to a drastic decrease in the embryonic cell number and blastocyst formation as compared toother farm animal species [1,2]. This raises several questions about whether the effects are endogenous or lack exogenous supportive signals. Preimplantation embryos are more competent when co-cultured with other embryos or maternal cells due to the production of paracrine or juxtracrine factors that interact to support the inefficient culture conditions associated with individually cultured embryos [3,4,5]. The cell number of in vivo–derived embryos is twice that of those that are generated in vitro, and the oviduct significantly affects the cell number and results in a 1.5-fold increase in cell number and hatching rates [6]. It shows that exogenous maternal factors are important for acquiring embryo developmental competence, and there are some other endogenous factors from the embryos themselves that may hamper the development of the embryo. Recent transcriptomics studies have shown vast differences between the porcine blastocysts that are produced in vivo and in vitro and demonstrate that upregulated gene expression of metabolism and arginine transporter contribute to the low developmental competence in in vitro–derived embryos [7].

To reveal the possible factors that regulate the blastocyst development and the rate of attachment in porcine embryos, we designed experiments to investigate the molecular impact of exogenous and endogenous signals responsible for embryonic–maternal crosstalk. For instance, the endogenous factors are formed or released by the embryos themselves, while the exogenous factors are released by the maternal cells and hamper the developmental competence such as miRNAs. Several cargos of protein, mRNA, miRNA, and metabolites are carried through the extracellular vesicles (EVs) and affect the growth of the embryo [4,8,9]. Moreover, the interplay between the EVs derived from the endometrium during embryo implantation in humans and animals has been investigated [10,11,12]. The porcine endogenous retrovirus (PERV) is secreted by all porcine cells and is considered a natural inhabitant of cells and biological fluids including the uterine cells. The endogenous retroviruses establish interplay between maternal and embryonic cells and are present in the exosomes released by the endometrium [13,14,15]. Studies also revealed a supportive role of Rho-associated coiled-coil-containing kinases (ROCK) in the development of cleaved embryos, while ROCK inhibition is critical during embryonic and pluripotent stem cell development [16,17,18], particularly trophoblast adhesion and differentiation [19,20].

The mechanism behind this interplay between the endogenous and exogenous factors that affect porcine embryo developmental competence remains unclear. Therefore, our study is an attempt to understand the interplay between the exogenous factors represented in endometrial EVs and the endogenous factors represented in PERV and ROCK pathways in the developmental competence of porcine embryos to enhance the production of more competent embryos that can meet the needs of cloning and xenotransplantation.

## 2. Materials and Methods

### 2.1. Chemicals

Unless otherwise specified, chemicals and reagents were purchased from Sigma-Aldrich (St. Louis, MO, USA).

### 2.2. Generation of Porcine Parthenogenetic Embryos

Porcine embryos were obtained through chemical parthenogenetic activation of in vitro matured oocytes as per our previous reports [5,21,22]. Porcine ovaries were collected from a slaughterhouse and transferred to the laboratory within 4 h in saline (NaCl 0.9%) at 30 °C. Cumulus–oocyte complexes (COCs) were retrieved through aspiration by an 18-gauge needle connected with a 10 mL syringe. Oocytes surrounded by compact layers of cumulus cells were selected using a stereomicroscope (SMZ 745T, Nikon, Tokyo, Japan) and washed three times in HEPES buffered Tyrode’s medium comprising 0.05% polyvinyl alcohol (TLH-PVA). COCs were cultured in 4-well dishes (Nunc, ThermoFisher Scientific, Roskilde, Denmark) containing 500 mL of a maturation medium comprising TCM-199 (Gibco, Waltham, MA, USA), 10% (*v/v*) porcine follicular fluid, cysteine (0.6 mM), sodium pyruvate (0.91 mM), epidermal growth factor (10 ng/mL), kanamycin (75 μg/mL), insulin (1 μg/mL), human chorionic gonadotrophin (10 IU/mL; Daesung Microbiological Labs; Uiwang, Korea), and equine chorionic gonadotrophin (10 IU/mL; Daesung Microbiological Labs) for 22 h. Then, the COCs were moved to the same culture conditions without the presence of the hormones for 22 h. Matured COCs were harvested, and cumulus cells were detached by gentle pipetting in hyaluronidase (0.6%) and then were washed in TLH-PVA and equilibrated in a pulsing medium consisting of mannitol (0.28 M), CaCl2 (0.1 mM), HEPES (0.5 mM), and MgSO4 (0.1 mM). Oocytes were then activated with a single direct current pulse of 1.5 kV/cm for 60 μs generated inside a glass chamber of two electrodes in an activation medium. The electric current was generated through a BTX Electro-Cell Manipulator 2001 (BTX Inc., San Diego, CA, USA). Activated oocytes were washed in TLH-PVA and cultured for 7 days in microdrops of porcine zygote medium-5 (PZM-5, Functional Peptides Research Institute Co. Ltd. (IFP), Yamagata, Japan) overlaid with mineral oil in a humidified atmosphere at 38.5 °C (5% O_2_, 5% CO_2_, and 90% N_2_). Blastocysts were obtained and washed in PBS and zona pellucida was removed by 0.1% pronase (*w/v* in PBS) to obtain zona-free embryos ready for further experiments.

### 2.3. Endometrium Culture

Uterine tissues of diestrus multiparous sows were collected from the slaughterhouse and transported to the lab within 4 h. Endometrium was separated aseptically under a laminar flow hood [23]. Endometrium was chopped into 1 mm pieces and seeded on 100 mm tissue culture dishes with a minimal volume of culture medium that comprised DMEM, 10% fetal bovine serum, and penicillin/streptomycin (100 U/mL penicillin, 100 µg/mL streptomycin) at 38.5 °C in a humidified atmosphere of 5% CO_2_. Tissue attachment and primary cell outgrowths were observed on day-5 of culture and the culture medium was then changed to a fresh one. Primary culture monolayer was maintained until day-8, and the tissue remnants were mechanically discarded.

### 2.4. Extracellular Vesicles Isolation and Characterization

On day-8, endometrial cell monolayers were cultured in a serum-free culture medium for 24 h and the conditioned medium was aspirated and centrifuged at 300× *g* to discard cell debris pellets [24]. EVs were isolated through targeted protein binding and nanofiltration using PureExo Exosomes Isolation kits (101 Bio, Palo Alto, CA, USA) [25] to yield 100 µL of EVs in phosphate-buffered saline (PBS) solution. EVs were characterized through ZetaView PMX 110 (Particle Metrix, Meerbusch, Germany) nanoflow fluorescence cytometry and nanoparticle tracking analysis instrument associated with ZetaView 8.05.14 SP7 software and Microsoft Excel 365 (Microsoft Corp., Seattle, WA, USA) [26]. After calibration with 100 nm polystyrene particles (ThermoFisher Scientific), one mL of the sample (diluted 20X in 1× PBS) was loaded into the machine and eleven different positions and two reading cycles per position were automatically set to measure the mean, median, and mode sizes (indicated as diameter), concentrations, and outlier removal in each sample. EVs were examined through transmission electron microscopy (TEM) [4,27]. In brief, 4 μL of isolated EVs solution was stained with 2% uranyl acetate, mounted on the center of 200-mesh copper grids, dried, and visualized through an OMEGA-energy filtering TEM (ZEISS LEO 912, Carl Zeiss, Jena, Germany) at 120 kV. The EVs cargo contents of some selected mRNAs, miRNAs, and proteins were analyzed through reverse-transcription polymerase chain reaction and proteomics as discussed below.

### 2.5. Embryo Attachment Model

Embryo attachment in feeder-free culture condition was established according to our previous method [28] with some modifications. Fifty μL microdrops of Matrigel basement membrane matrix (BD Biosciences, San Jose, CA, USA) were placed on 4-well dishes (Nunc) and incubated for 30 min at 38.5 °C. Matrigel was removed and replaced with 50 μL of culture medium that was composed of DMEM/F-12 supplemented with 10% fetal bovine serum, β-mercaptoethanol (0.1 mM), 1% nonessential amino acids (Invitrogen, Waltham, MA, USA), and 1% penicillin/streptomycin (100 U/mL penicillin, 100 µg/mL streptomycin). The microdrops were covered with mineral oil and incubated for 30 min before embryo placement. On day-7, embryos were collected and zona pellucidae were removed by pronase (0.1% in PBS) for 1 min at 38.5 °C. Embryos were washed with the culture medium before placing them into the Matrigel-coated microdrops. Embryos were then incubated in a humidified atmosphere of 5% CO_2_ at 38.5 °C and monitored for attachment and outgrowths on days 2–5 from culture.

### 2.6. Experimental Design

#### 2.6.1. Effects of Endometrial-EVs and ROCK Inhibitor on Embryo Development and Attachment

First, embryos (*n* = 20 for 3 replicates) were divided into 4 groups: control, 10 µM ROCK (Rho-associated coiled-coil containing kinases) inhibitor (RI, Y-27632) [29], Endo-EVs (1.5 × 10^7^ particles/mL) [25,30], or combined supplementation of RI and Endo-EVs for different durations (i.e., 36 h and extended to 5 days). The control group was cultured in a plain culture medium without supplementation. Embryos were monitored for attachment, cell number, apoptosis, outgrowths, immunofluorescence staining of pluripotency marker (Oct4) and trophoblast marker (Cdx2), and relative quantitation of some miRNA and mRNA transcripts’ expression that are related to apoptosis, cell attachment, cell cycle, and embryo development.

#### 2.6.2. Effect of miR-155 on Embryo Development and Attachment

Based on the findings of EVs analysis, we designed experiments to explore the roles of miR-155 in maternal-embryonic communications. The mir-155 inhibitor was supplied to examine its effect on Endo-EVs supplementation on embryonic development and attachment. Moreover, the effects of miR-155 mimic on embryonic development and attachment were studied in combination with or without RI.

#### 2.6.3. Effect of Targeting PERV on Embryo Development and Attachment

Based on the findings of EVs analysis, we targeted PERV with CRISPR/Cas9 to explore the role and impact of EVs derived from PERV-depleted endometrium on embryo development and attachment.

### 2.7. EVs Labeling and Uptake

Before EVs isolation, a serum-free conditioned medium was mixed with the PKH26 lipophilic fluorescent stain (Invitrogen) according to the manufacturer’s instructions, and EVs were isolated to remove the excess free PKH67 dye following the manufacturer’s recommendations [31,32]. EVs were then supplemented (1.5 × 10^7^ particles/mL) [25,30] with cultured embryos for 24 h to monitor their uptake through a fluorescent microscope (MshOt, Guangzhou Micro-shot Technology Co., Ltd., Guangzhou, China). For negative control staining, the plain conditioned medium was mixed with PKH26 and processed by the same EV labeling procedure.

### 2.8. Immunofluorescence

Immunofluorescence staining of OCT4 and CDX2 was performed according to our previous protocol [33] with some modifications as follows: attached embryos on day-5 were fixed in 4% paraformaldehyde (*w/v* in PBS), pH 7.4 for 15 min at room temperature. Fixed embryos were washed in PBS, permeabilized with 0.1% Triton-X100 (*v/v* in PBS) for 10 min, and then were blocked by 1% goat serum (*v/v*; Invitrogen) for 30 min at room temperature. Primary antibodies specified against Oct4 (mouse monoclonal IgG2b, sc-5279, Santa Cruz Biotech. Inc., Santa Cruz, CA, USA) and Cdx2 (rabbit monoclonal IgG, ab76541, Abcam, Seoul, Korea) were diluted (1:100) and prepared in PBS. The attached embryos were incubated with the primary antibodies (1 h at 38.5 °C), washed in PBS three times, then incubated with the secondary antibodies (Alexa Fluor 488 goat anti-mouse IgG, A11001 and Alexa Fluor 568 goat anti-rabbit IgG, A11011; Invitrogen, Life Technologies Corp., Eugene, OR, USA), and the resulting solution was diluted (1:200) and kept in PBS for 1 h at 38.5 °C before washing in PBS three times. Embryonic cell nuclei were counterstained with Vectashield antifade mounting medium containing 40,60 -diamidino-2-phenylindole (DAPI; Vector Laboratories, Vector Laboratories, Burlingame, CA, USA) for 5 min, and the fluorescence signals were examined with a fluorescent microscope at 488 nm, and 568 nm for Oct4 and Cdx2, respectively. Images were captured and the fluorescence intensity pixel analysis was analyzed with ImageJ 1.53k software (National Institute of Health, Bethesda, MD, USA).

### 2.9. TdT-Mediated dUTP-X Nick End Labeling (TUNEL) Assay

Labeling of DNA strand breaks and detection of apoptotic cells were examined through In Situ Cell Death Detection TUNEL assay Kit, Fluorescein (Roche Holding AG, Basel, Switzerland) according to the manufacturer’s protocol. Embryos were fixed in 4% paraformaldehyde and permeabilized in 0.1% TritonX and then incubated with the working solution of an enzyme (TdT) and a label solution (fluorescein-dUTP) for 1 h at 38.5 °C. Nuclei were counterstained with Vectashield antifade mounting medium as mentioned above. Green fluorescence positive cells (apoptotic cells) were captured and counted with ImageJ software.

### 2.10. MiR-155 Mimic, miR-155 Inhibitor, and CRISPR/Cas9 Transfection

We used Campylobacter jejuni CRISPR/Cas9 (cjCas9) vector to cleave PERV mRNA. We cloned cjCas9-based sgRNA targeting env gene of PERV in our cjCas9 vector with slight modification (D8A for inactivating RuvC domain) (Appendix A). CRISPR/Cas9 vector (1 mg), miR-155 mimic (100 nM), and miR-155 inhibitor (100 nM) oligonucleotide sequences (Table 1) [34] were transfected to the embryos [35] with some modifications. The nucleic acids were incubated with Lipidofect-P transfection reagent (Cat # LDL-P001, Lipidomia, Gachon University IT Center, Gyeonggi-do, Korea) for 30 min at room temperature, and then the mixture was supplemented to the embryo culture medium and incubated for the attachment and further development.

### 2.11. Conventional and Real-Time Polymerase Chain Reaction

Total RNA was extracted from the embryos (*n* = 5, 4 replicates) using RNeasy Micro Kit (Qiagen GmbH, Hilden, Germany, Cat #74004) that included DNase I for removing any of DNA residuals. NanoDrop 2000 (Thermo Scientific) was used to determine the quality of the extracted RNA. Values of > 1.8 of OD 260/280 and 260/230 ratios were used for the reverse transcription. Complementary DNA (cDNA) was synthesized using 2X RT Pre-Mix of QuantiNova Reverse Transcription Kit (Qiagen) with a total volume of 20 μL (1 μg of total RNA, 4 μL of 5× RT buffer, 1 μL of reverse transcriptase enzyme mix, 1 μL of oligo dT primer for mRNA or universal stem-loop primer for miRNAs (Table 1), and RNase-free distilled water to reach the volume of 20 μL). Pulsed reverse transcription was conducted to generate complementary DNA (cDNA). Twenty nanograms of total RNA in a 20 μL reaction volume was used as a template for cDNA synthesis in 60 cycles of 2 min at 16 °C, 1 min at 37 °C, and 0.1 s at 50 °C, and a final inactivation at 85 °C for 5 min [36,37]. For conventional PCR, cDNA was amplified by using 2X Taq PCR Pre-Mix (BioFACT, Seoul, Korea) in the following conditions: initial denaturation (2 min at 95 °C), 40 amplification cycles of denaturation (95 °C for 20 s), annealing (60 °C for 30 s), and extension (72 °C for 30 s), followed by a final extension step of 5 min at 72 °C. The PCR products were analyzed by a Gel Doc XR+ UV transilluminator with Image Lab Software (Bio-Rad, Berkeley, CA, USA) on a 2.5% agarose gel (Amresco, Cleveland, OH, USA) stained with ethidium bromide (BioFACT). Gel electrophoresis was performed using Mupid^®^-One (TAKARA, Tokyo, Japan) at 135 V for 25 min. Relative quantitative PCR was performed using the CFX Connect Real-Time PCR system (Bio-Rad) and SYBR 2X Real-Time PCR Pre-Mix (BioFACT). Details about the target genes, housekeeping mRNA and snRNA, primers, and product size are listed in Table 2. The reaction mixture (20 μL) comprised 10 μL of SYBR^®^ Premix (2×), 2 μL of cDNA (100 ng), 2 μL of forward and reverse primers (10 μM), and 6 μL of distilled water. Cycling conditions were 95 °C (1 min) followed by 40 PCR cycles of 95 °C (5 s, DNA denaturation), 60 °C (30 s, primer annealing), and 72 °C (30 s, extension). Primer specificity was determined by melting curve protocol ranging from 65 to 95 °C and was confirmed with single peaks in the melt curves, gel electrophoresis, and cDNA-exempted samples. Transcripts of the target genes were compared to those of housekeeping genes (GAPDH-mRNA and U6-snRNA). The gene networks of these studied transcripts were analyzed through GeneMANIA webtool (https://genemania.org/, accessed on 30 September 2022) [21].

### 2.12. Preparation of Endo-EVs Protein Fraction, In-Gel Digestion and Proteomic Analysis by LC–MS/MS

The protocol was performed according to our recent report [27]. In brief, EVs pellets were suspended and dialyzed against 10 volumes of 20 mM Tris-HCl (pH 8.0) (the molecular mass cutoff of was 10,000 Da). Protein concentration was estimated through the bicinchoninic acid method and then proteins were fractionated by sodium dodecyl sulfate-polyacrylamide gel electrophoresis (SDS-PAGE). For Coomassie Brilliant Blue staining, the gels were destained by 50% acetonitrile and 10 mM ammonium bicarbonate solution [46] and then gels were rinsed twice with distilled water, followed by 100% acetonitrile, respectively and then dried with a speed vacuum concentrator. The gels were treated with mixture of 10 mM dithiothreitol and 100 mM ammonium bicarbonate at 56 °C, before treatment with 100 nM iodoacetamide to minimize alkylate S–S bridges. The gels were vortexed in three volumes of distilled water for washing and then dried with a speed vacuum concentrator. The gels were incubated in 50 mM ammonium bicarbonate and 10 ng/mL trypsin at 37 °C for 12–16 h for tryptic digestion. Tryptic peptides were retrieved after treatment with 50 mM ammonium bicarbonate and 50% acetonitrile containing 5% trifluoroacetic acid. Peptide extract was lyophilized and stored at 4 °C until further analysis. Tryptic peptide extract was suspended in 0.5% trifluoroacetic acid and 10 μL from each sample was loaded onto MGU30-C18 trapping columns (LC Packings) to concentrate peptides and clear extra chemicals. Concentrated tryptic peptides were eluted from the column and loaded onto a 10 cm × 75 μm I.D. C18 reverse-phase column (PROXEON, Odense, Denmark) at adjusted flow rate (300 nL/min). Peptides were retrieved by a gradient of 0–65% acetonitrile for 80 min. MS and MS/MS spectrum was obtained by using LTQ-Velos ESI ion trap mass spectrometer (Thermo Scientific, Waltham, MA, USA). MASCOT 2.4 was used to analyze MS/MS data with a false discovery rate of 1% as a cutoff value. Protein quantities were estimated through the exponentially modified protein abundance index (emPAI) and were expressed as mol %. Three technical replicates were performed. Functional analysis and gene ontology were performed through the Functional Annotation Tool, DAVID Bioinformatics Resources (NIAID/NIH; https://david.ncifcrf.gov/home.jsp, accessed on 16 August 2022) [36,47].

### 2.13. Statistical Analysis

For each experiment, an average of 20 embryos and at least 3 replicates were used for the analysis. Lieven’s test and Kolmogorov–Smirnov test were used to confirm the homogeneity of variance and the normality of distribution, respectively. Data were expressed as mean ± standard error of means (SEM) or standard deviation (SD) and examined through using an unpaired Student *t*-test or with univariate analysis of variance (ANOVA) followed by Tukey’s multiple comparison test, respectively. Statistical significance was considered at *p* <0.05 or *p* <0.01. GraphPad Prism 5 (GraphPad Software Inc., San Diego, CA, USA) was used for the statistical analyses.

## 3. Results

### 3.1. Endo-EVs Isolation, Characteristics, and Cargo Contents

Endometrial cells were successfully cultured with the characteristic flattened epithelial properties and were maintained in culture until day-8 (Figure 1A,B). Cells were cultured in a serum-free culture medium for 24 h to retrieve the Endo-EVs. ZetaView analysis showed a presence of 1.1 × 108 particles/mL of average particle size 115.6 ± 28.4 nm in the isolated conditioned medium (Figure 1C). Furthermore, TEM images revealed the presence of lipid bilayer vesicles (Figure 1D) that characterize the appearance of EVs. The isolated Endo-EVs were analyzed with qPCR and showed the expression of certain miRNAs and mRNAs when compared to those of corresponding endometrial origin (Figure 2). Endo-EVs contained miR-100, miR-132, and miR-155, β-catenin, and PERV, but we could not observe GAPDH mRNA in the isolated EVs. Endo-EVs were further characterized through the profiling of protein contents by proteomics. We identified 82 proteins in the Endo-EVs (Appendix A), of which the top 20 proteins that constituted around 65% of the total proteins of the Endo-EVs included proteins associated with cytoskeleton structures (e.g., keratins), extracellular matrix (e.g., plasminogen), and calcium metabolism (e.g., vitamin-D-binding protein, and calcium-binding protein A2); however, they contain apoptosis-related proteins such as procathepsin (Table 3). Detailed information about the proteins and their functions are listed in Appendix A.

### 3.2. Effect of Endo-EVs and ROCK-Inhibitor (RI) on Embryo Attachment and Outgrowths

The embryonic uptake of Endo-EVs was confirmed by the presence of intracytoplasmic fluorescence signals after labeling Endo-EVs with PKH26 stain and their incubation with the embryos for 30 h (Figure 3).

We then investigated the embryonic attachment and the development after supplementing the culture medium with EVs or with RI or with their combination together. When compared with the control group, Endo-EVs supplementation for 36 h showed significant improvement in embryo attachment (65.55% vs. 34.43%, *p* < 0.01), increased embryonic cell number (26 vs. 21.8, *p* < 0.01), and a significant reduction in apoptosis (2.37% vs. 26.1%, *p* < 0.01) (Figure 4A–D). Based on these preliminary experiments, we found that RI can reduce apoptosis in embryonic stem cells (Figure 4D); however, it could not support the embryonic development (Figure 4C) compared to control and Endo-EVs groups. Therefore, we examined the beneficial effects of both Endo-EVs and RI to support the embryonic development, which showed 100% embryonic attachment with a significant increase in cell numbers (mean = 33.6) and a significant reduction in the ratio of apoptotic cells (1.57%) when compared to those of the experimental groups (Figure 4A–D).

We followed up the development of embryos supplanted with Endo-EVs and RI for 5 subsequent days and compared that with the control group. The results showed a significant increase in the embryonic outgrowths on day-5 in the embryos supplemented with Endo-EVs and RI (72.9% vs. 32%) (Figure 5A,B).

Furthermore, the expression of Oct4 and Cdx2 in the Endo-EVs and RI-treated embryonic cells was significantly increased by 2.45-fold and 3.48-fold, respectively, compared to those of the control group (Figure 6A–C).

Additionally, the qPCR analysis showed that combined supplementation of Endo-EVs with RI significantly reduced the expression of Bax (0.6-fold) and miR-155 (0.17-fold) but increased the expression of Bcl2 (4.73-fold), Cdk2 (4.33-fold), PERV (2.55-fold), β-catenin (7.13), interferon-gamma (1.7-fold), Zeb1 (1.9-fold), PTN mRNAs (9.83-fold), miR-100 (3.79-fold), and miR-132 (16.1-fold) compared to those of the control group (Figure 7).

### 3.3. Impact of miR-155 on Embryo Attachment and Development

Based on the previous results, we speculated that the miR-155 contents of Endo-EVs could exert a negative impact on embryo attachment, and therefore we specifically targeted miR-155 with an inhibitor (miR-155 inhibitor). Treatment of embryos with Endo-EVs and miR-155 inhibitor significantly improved the attachment (90% vs. 50%, *p* < 0.01), increased the cell number (30 vs. 12, *p* < 0.01) but had no effects on the apoptosis ratio (2% vs. 3.16%, *p* = 0.27) as compared to those of Endo-EVs supplemented group (Figure 8A–D).

Additionally, individual treatment with miR-155 mimic showed a significant reduction (*p* < 0.05) in the attachment (20.8%) and cell number (*n* = 6) and a significantly increased apoptosis ratio (48.3%). Moreover, these effects were significantly alleviated with an individual treatment of RI (45.6%, 14, and 14.66% for attachment ratio, cell number, and apoptosis ratio, respectively; *p* < 0.01) (Figure 9A–D). Hence, we speculated that RI could antagonize the negative impact of miR-155 on embryonic attachment and development.

### 3.4. Effects of Endo-EVs PERV Depletion by CRISPR/Cas9 on Embryo Attachment and Development

Furthermore, based on the Endo-EVs cargo contents of PERV, we targeted PERV with CRISPR/Cas9 to examine the effects of PERV reduction on embryonic attachment and development. The designed CRISP/Cas9 was successfully transfected and expressed in the cells as indicated by the green fluorescence protein expression in Figure 10A,A’,A’’. PERV expression was significantly reduced (0.23-fold, *p* < 0.05) compared to that of control endometrium cells. Surprisingly, miR-155 expression showed a 6.16-fold increase (*p* < 0.05) in PERV-depleted endometrium compared with that of the control endometrium. Similarly, the derived EVs from PERV-depleted cells showed a significant reduction in PERV mRNA expression (0.27-fold, *p* < 0.05), and a significant increase in miR-155 (4.9-fold, *p* < 0.01). Moreover, supplementation of embryos with EVs from PERV-depleted cells significantly reduced the attachment and day-5 outgrowth ratios compared to those of control EVs (49% vs. 65.8% and 18% vs. 31%, respectively, *p* < 0.05).

## 4. Discussion

In this study, as a continuation of our work [28], we generated a model of culturing porcine embryos in feeder-free culture conditions using Matrigel basement membrane matrix but on a microdrop level. This model achieved 100% of blastocyst attachment and embryonic outgrowths with the help of Endo-EVs supplementation and ROCK pathway inhibition.

Recently, the interplay between EVs derived from the endometrium during the embryo implantation in humans has been investigated [10,11,12]. Our results showed that Endo-EVs enhanced embryonic attachment and adhesion to the Matrigel basement membrane matrix, which is in accordance with previous studies [11,48,49,50,51]. This improvement may be attributed to the transfer of proteins associated with cell attachment, cytoskeleton integrity, calcium metabolism as revealed by proteomics analysis. Additionally, embryonic development was improved because of β-catenin transfer through the Endo-EVs cargo that increased the expression of β-catenin in embryos. β-catenin plays a crucial role in endometrium functions and is considered an imperative signal in invasion and differentiation of trophoblasts and embryo implantation [39,52]. Moreover, Endo-EVs cargo contained miR-100, which is expressed in human endometrial cell-derived EVs and activates focal adhesion kinase (FAK) and c-Jun N-terminal kinase (JNK) and promotes the invasion and migration of human and goat trophoblasts [12,53,54]. Furthermore, Endo-EVs contained miR-132 that is expressed temporally in porcine endometrium at the time of embryo implantation [55] and is a potential factor for enhancing trophoblast invasion and embryo implantation by targeting death-associated protein kinase 1 (DAPK-1) [56].

In our study, several mRNA transcripts (antiapoptotic gene (BCL2), zinc finger E-box-binding homeobox 1 (Zeb1), β-catenin, interferon-γ (IFNG), protein tyrosine phosphatase non-receptor type 1 (PTPN1), and cyclin-dependent kinase 2 (CDK2)) were increased in the embryonic cells after EVs supplementation. Moreover, the pluripotency master Oct4 and the trophoblast associated gene CDX2 were also increased in the EVs supplementation which might be attributed to embryo developmental competence observed in this group [4,28,57,58,59]. These genes are of important roles in embryonic development, implantation, trophoblast attachment, and stem cell growth, as well as the cell cycle and survival as we revealed in our previous reports [5,21]. Furthermore, bioinformatics analysis indicated an existing physical interaction, shared protein domains, common pathways, and co-expression of the studies genes (Appendix A). Detailed functions of the genes are listed in Table 2.

On the other hand, Endo-EVs contained miR-155, which inhibits trophoblast migration and proliferation [60,61], increases preeclampsia in patients, and induces trophoblast apoptosis by targeting BCL2 (apoptosis regulator) [62]. Furthermore, some of the cargo proteins in the Endo-EVs included proapoptotic signals such as cathepsin and procathepsin. Paradoxically, miR-155 showed a 1.6-fold increase in the mouse uterus during the receptive phase of embryo attachment, which suggests a modulatory role of miR-155 during this critical stage in mice [63].

In this study, the ROCK inhibitor (Y-27632) abolished all negative impacts of miR-155 on embryo attachment and development. RI remarkably reduces the tumor necrosis factor (TNFα)-induced upregulation of miR-155 [64]. RI interferes with the cargo transfer from microparticles and impairs their ability to mediate extracellular signaling [65]. Therefore, we speculate the likelihood of other mechanisms that are associated with miRNA export upstream to RhoA/ROCK signaling. Another mechanism is the antiapoptotic action of RI can antagonize the apoptotic action of miR-155 on embryonic cells [18,66,67,68,69]. Additionally, ROCK pathway inhibition enhances trophoblast adhesion and viability in humans. Paradoxically, RI can reduce the trophoblast migration of human extravillous trophoblasts [70]. This is in accordance with our findings regarding the ameliorative effects of RI on the negative impacts of both individual and EVs-transmitted miR-155; however, individual RI improved the embryonic attachment and development. Therefore, RI synergize the actions of Endo-EVs through antagonizing the effects of the non-useful cargo contents of Endo-EVs, such as miR-155.

Computational analysis of miR-155 targets (http://mirdb.org/, accessed on 16 August 2022) showed that they interfere with a cell-cycle-related gene (CDK2-associated cullin domain 1 (CACUL1)) (target score 82%) and an antiapoptotic gene (BCL2-associated athanogene 5 (BAG5)) (target score 70%), which had correlation with other proteins involved in cell apoptosis and growth, including BCL-2. Our qPCR data showed that RI ameliorates the negative effects on mRNA expression of CDK2 and BCL2, which could help the cell cycle and reduce apoptosis. Moreover, miR-155 targets catenin alpha 3 (CTNNA3) (target score 67%) which belongs to the catenin family and encodes a protein that plays a role in cell-to-cell adhesion. Similar findings in qPCR have been shown in β-catenin expression. Furthermore, miR-155 targets protein tyrosine phosphatase, non-receptor type 2 (PTPN2) (target score 84%), which regulates various cellular processes including cell growth, differentiation, and mitotic cycle. Moreover, we found positive effects of RI on the expression of PTPN1. Therefore, we inferred that miR-155 can interfere with several essential pathways related to cell growth and differentiation and cause apoptosis.

The EVs cargo contained PERV mRNA, which coincides with some recent reports showing exosomes that contain mRNAs for ovine endogenous jaagsiekte retroviruses (enJSRV-ENV) [13] and human endogenous retroviruses [71]. There is a consensus about the essential roles of endogenous retroviruses in the early stages of embryo attachment physiological functions of trophoblasts and placentation [15,72,73,74,75]; however, the role of PERV in porcine embryo attachment remains unclear. A recent report showed that targeting PERV with CRISPR/Cas9 at the zygote stage impaired the blastocyst development and indicated the essential roles of PERV for the preimplantation embryonic development [14]. Our results indicate that PERV targeting in EVs could have reduced the embryonic attachment and development. This reduction might be due to the increased levels of miR-155 in the transferred cargo contents of EVs. The reason behind these increased levels is unclear and might be attributed to the essential roles of PERV in cellular viability, normal physiological functions, and the indel mutations caused by CRISPR/Cas9 [14]. Furthermore, avian endogenous retrovirus shows negative regulation with miR-155, which is suggestive of the interplay between ERVs and miR-155 [76]. The human and zebrafish microRNA-155 target the corresponding HERV and ZFERV env sequence, which indicates that miR-155 targeting ERVs env is mostly conserved in animals and may regulate ERVs activity [76]. We speculate that EVs can carry both useful and harmful cargo contents and ameliorating the de trop cargo contents (such as miR-155) can maximize the useful effects of EVs, especially during embryo implantation and maternal recognition of pregnancy.

## 5. Conclusions

To our knowledge, this is the first attempt to understand the roles of EVs cargo in determining embryo developmental competence and mediating molecular signaling between the embryo and the endometrium in the pig. Endometrial EVs improved embryo attachment, increased cell numbers and reduced apoptosis, and the unwanted effects of their de trop cargo contents of miR-155 can be alleviated through anti-apoptotic molecules such as the ROCK inhibitor. This model would help in establishing an extended culture system to understand early embryonic stem cell differentiation. The current model would provide a paradigm for studying the embryonic–maternal crosstalk and to develop pharmaceutical criteria for improving pregnancy outcomes in porcine species.

## Figures and Tables

**Figure 1 cells-11-03178-f001:**
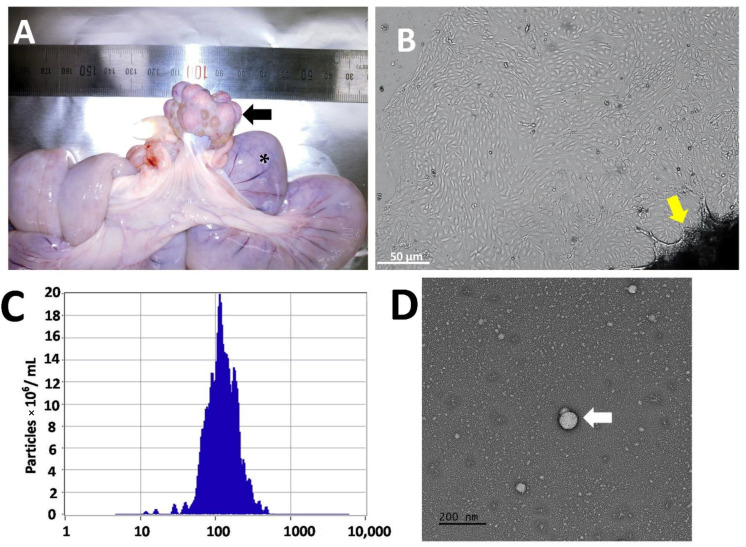
Obtaining the endometrial-derived extracellular vesicles (Endo-EVs). (**A**) Porcine endometrium (*n* = 6) was retrieved from the uterus (*) of diestrus sows (as indicated by corpora lutea, the black arrow); (**B**) primary endometrium cell culture was established (white arrow) from the endometrial tissue flakes (yellow arrow). On day-8 of primary outgrowths (Scale bar = 50 µm), the tissue chops were removed, and the cells were cultured in a serum-free culture medium to collect the conditioned medium for Endo-EVs isolation. (**C**) Endo-EVs were isolated by targeted nanofiltration method and were characterized by ZetaView nanoflow cytometry and nanoparticle tracking analysis and showed an average diameter of 115.6 ± 28.4 nm with a concentration of 1.1 × 10^8^ particles/mL (dilution factor is 20X). (**D**) Endo-EVs were visualized by transmission electron microscope and showed bilipid vesicles (white arrow).

**Figure 2 cells-11-03178-f002:**
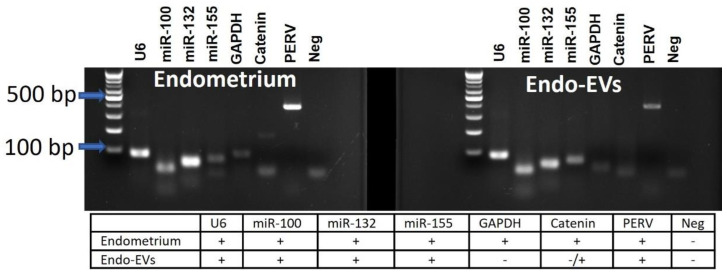
Images of gel electrophoresis of mRNA and miRNA of the endometrium and their derived extracellular vesicles (Endo-EVs). The PCR products were electrophorized in agarose gel (2%), and the bands were visualized using a 100 bp DNA ladder as reference. The band expression of the snRNA (U6) and GAPDH were used as housekeeping genes for miRNA and mRNA, respectively. We contrasted the expression in Endo-EVs and found that some mRNAs were not expressed in the Endo-EVs such as GAPDH and catenin. For more details about the PCR product size, please refer to Table 1 in Section 2.

**Figure 3 cells-11-03178-f003:**
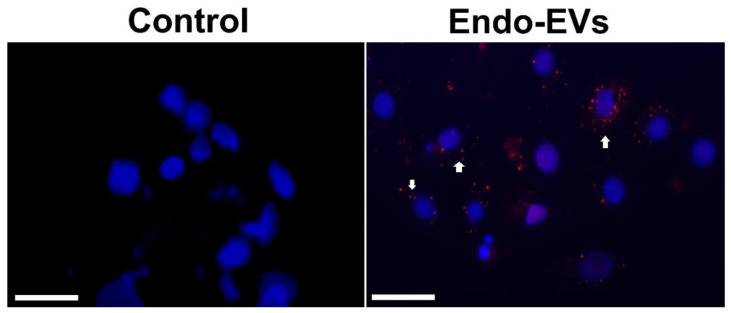
Cellular uptake of Endo-EVs. Endo-EVs were stained with a lipophilic live-imaging dye PKH26, and the free dye was removed by washing during isolation. Plain conditioned medium was processed in the same way as the isolated EVs and worked as negative control (Control). Attached embryonic cells were incubated with the stained control and Endo-EVs for 24 h and then were stained with DAPI and visualized by fluorescence microscope. White arrows indicate the presence of cytoplasmic stained EVs surrounding the nuclei. Scale bar = 20 µm.

**Figure 4 cells-11-03178-f004:**
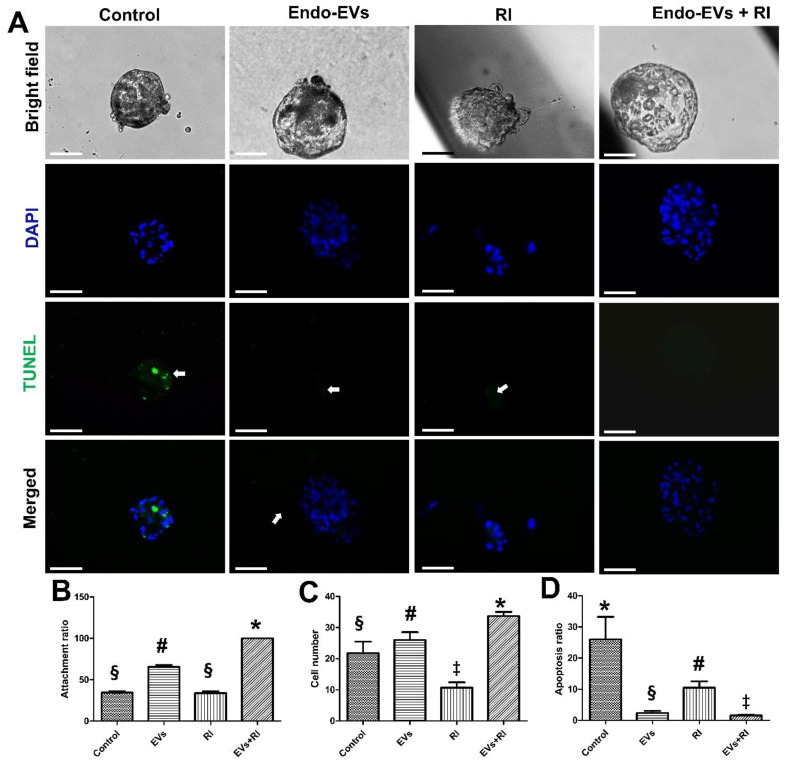
The effects of endometrial EVs (Endo-EVs) and ROCK inhibitor (RI) on porcine embryo development. (**A**) Day-7 zona-free embryos (*n* = 20, 3 replicates) were cultured on Matrigel-coated dishes in microdrops of culture medium in a humidified atmosphere of 5% CO_2_ for 36 h. The control group was cultured in a plain culture medium while the RI group in a medium supplemented with Y-27632 (10 µg/mL) and EVs group in a medium supplemented with Endo-EVs of 2.6 × 106 particles/mL. In the combined group, embryos were cultured in a medium supplemented with both RI and EVs of the same working concentrations. Scale bar = 100 µm. All groups were imaged in a bright field before staining with TUNEL assay and contrasted with DAPI stain. White arrows indicate the apoptotic cells; (**B**–**D**) graphs show the embryo attachment, cell number, and apoptosis among the groups, respectively. Values (mean ± SD) were compared with ANOVA followed by Tukey’s test to determine the difference among the groups. Values denoted by ‡, #, §, and * were considered statistically different (*p* < 0.05).

**Figure 5 cells-11-03178-f005:**
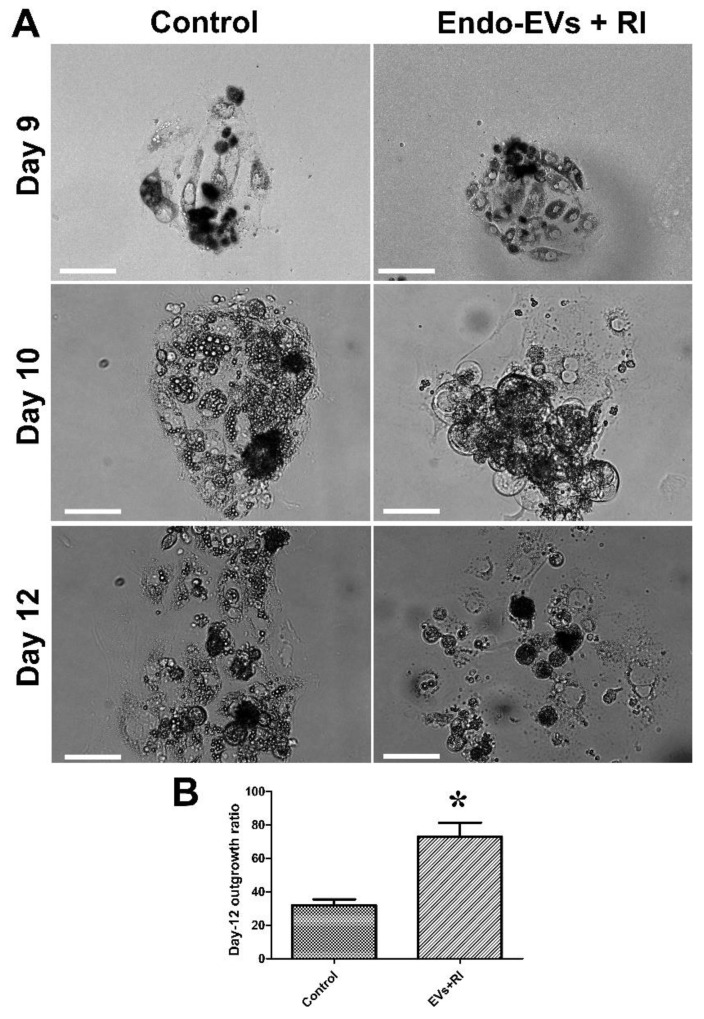
The effect of combined treatment of Endo-EVs and ROCK inhibitor (RI) on embryonic outgrowths. (**A**) Day-7 zona-free porcine embryos (*n* =18 for 3 replicates) were cultured on Matrigel-coated dishes in microdrops of culture medium in a humidified atmosphere of 5% CO_2_ for 5 days. Scale bar = 50 µm; (**B**) the averages of percentages of embryonic cell outgrowths were calculated on day-2, day-3, and day-5 (mean ± S.E.M.) and compared with Student’s *t*-test. Asterisk (*) indicates a significant difference (*p* < 0.05).

**Figure 6 cells-11-03178-f006:**
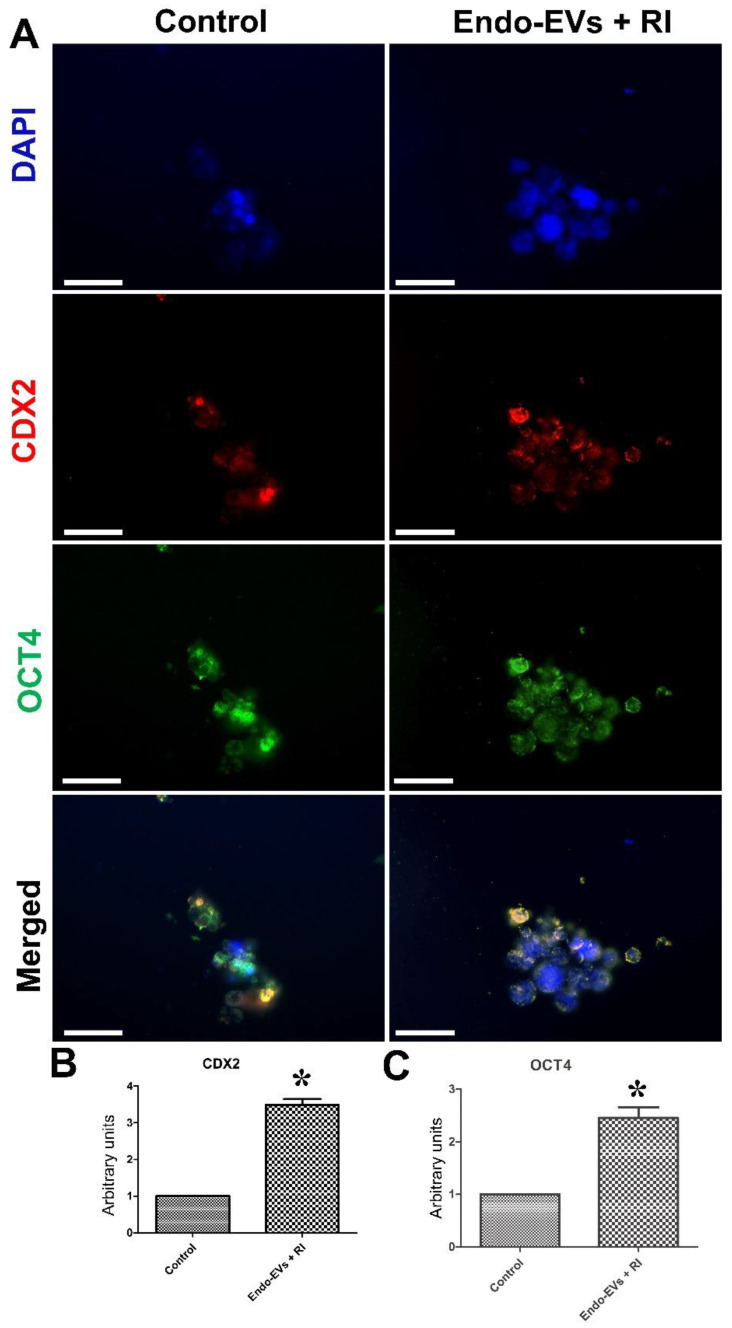
The effect of combined treatment of Endo-EVs with ROCK inhibitor (RI) on OCT4 and CDX2 expression in cultured embryonic cells. (**A**) Day-7 zona-free porcine embryos (*n* = 18 for 3 replicates) were cultured on Matrigel-coated dishes in microdrops of culture medium in a humidified atmosphere of 5% CO_2_ for 5 days and then incubated with primary antibodies specific to OCT4 and CDX2 followed by corresponding specific secondary antibodies. Scale bar = 50 µm. (**B**,**C**) Images of CDX2 and OCT4, respectively, were analyzed with ImageJ software to compare the pixels of the fluorescence intensity in the same exposure time, contrast, and area of analysis. The values were normalized to the control group as an arbitrary unit to show the fold of change between the groups. Values (mean ± S.E.M.) were analyzed with Student’s *t*-test. Asterisk (*) indicates a significant difference (*p* < 0.05).

**Figure 7 cells-11-03178-f007:**
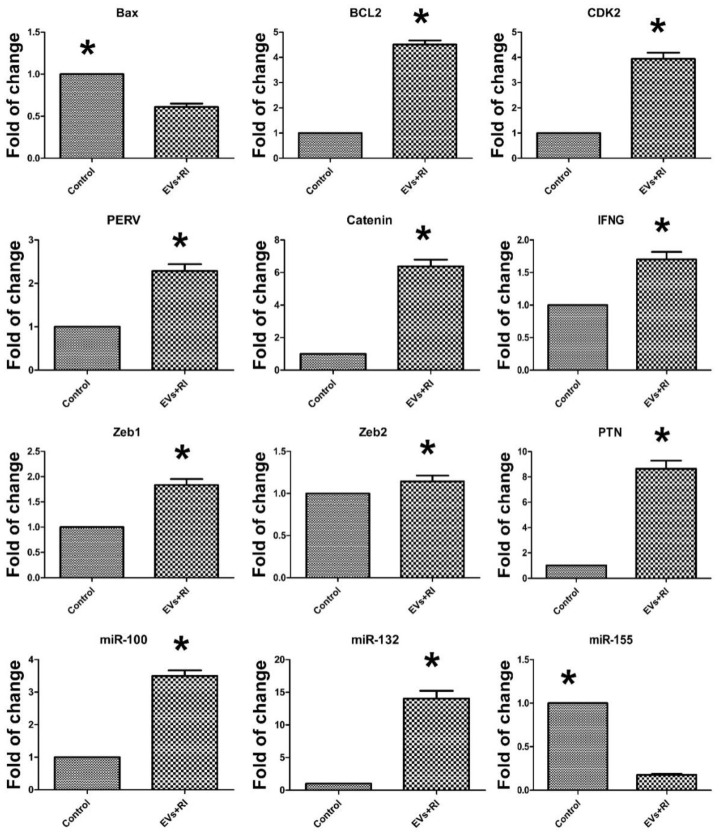
Relative quantitative analysis (RT-qPCR) of mRNA transcripts expressed in the embryos treated with Endo-EVs and RI. Five blastocysts from each group for 4 replicates were used for qPCR analysis. The means were normalized to the control group and expressed as arbitrary units. Data were expressed as mean ± S.E.M. and the difference between the two groups was compared with Student’s *t*-test. Values denoted by an asterisk (*) were considered statistically significant (*p* < 0.05).

**Figure 8 cells-11-03178-f008:**
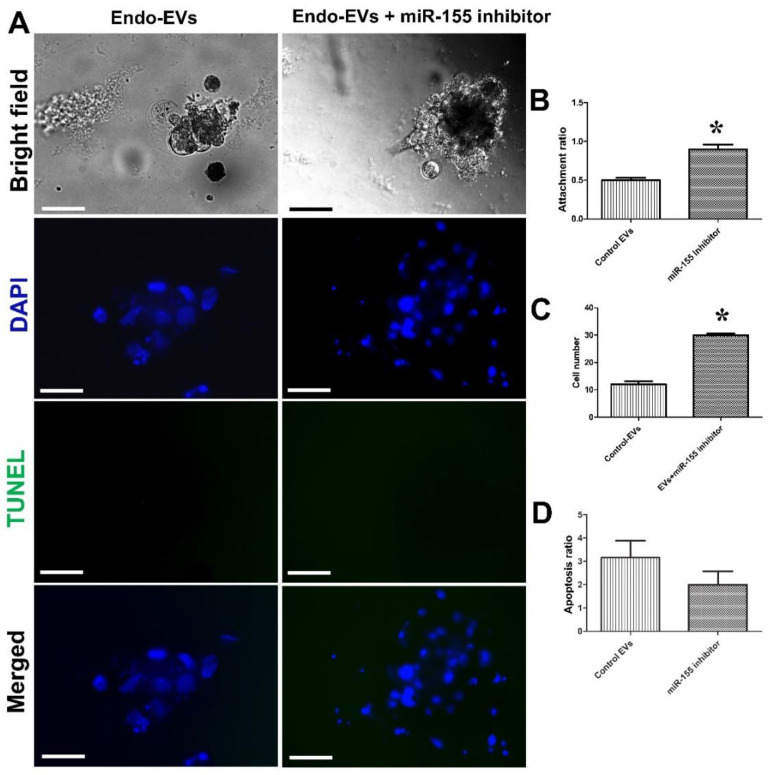
Investigating the effects of the miR-155 inhibitor on embryonic attachment and development. (**A**) MiR-155 inhibitor was designed (Table 1) and transfected to day-7 zona-free porcine embryos. Embryos were cultured on Matrigel-coated dishes in microdrops of culture medium containing Endo-EVs in a humidified atmosphere of 5% CO_2_ for 5 days. Control Endo-EVs group was treated the same as the miR-155 group except for the absence of RNA sequence during transfection. Scale bar = 50 µm. All groups were imaged in a bright field before staining with TUNEL assay and contrasted with DAPI stain; (**B**–**D**) graphs show the embryo attachment, cell number, and apoptosis among the groups, and values (mean ± SEM) were compared with Student’s *t*-test to determine the difference among the groups. Values denoted by an asterisk (*) were considered statistically significant (*p* < 0.05).

**Figure 9 cells-11-03178-f009:**
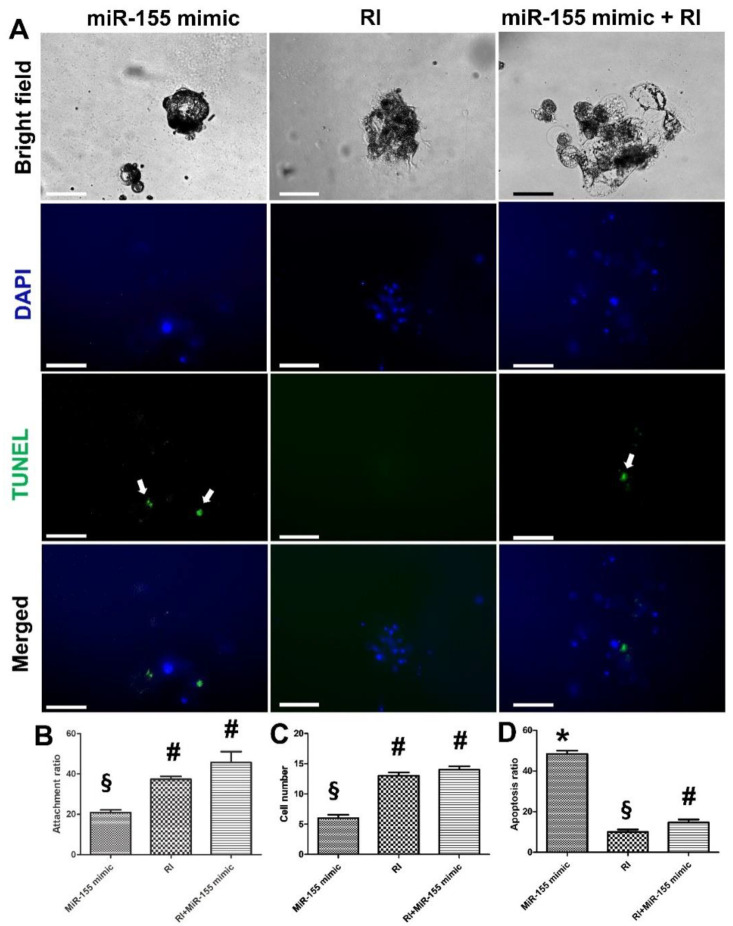
Investigating the effects of miR-155 mimic and ROCK inhibitor (RI) on embryonic attachment and development. (**A**) MiR-155 mimic duplex was designed (Table 1) and transfected to day-7 zona-free porcine embryos. Embryos were cultured on Matrigel-coated dishes in microdrops of culture medium in a humidified atmosphere of 5% CO_2_ for 5 days. The three groups were treated the same as the miR-155 mimic group except for the absence of RNA sequence during transfection of the RI group. Scale bar = 50 µm. All groups were imaged in a bright field before staining with TUNEL assay and contrasted with DAPI stain. White arrows indicate the apoptotic cells; (**B**–**D**) graphs show the embryo attachment, cell number, and apoptosis among the groups. Data were expressed as means ± S.E.M. Values (mean ± SD) were compared with ANOVA followed by Tukey’s test to determine the difference among the groups. Values denoted by symbols #, §, and * were considered statistically significant (*p* < 0.05).

**Figure 10 cells-11-03178-f010:**
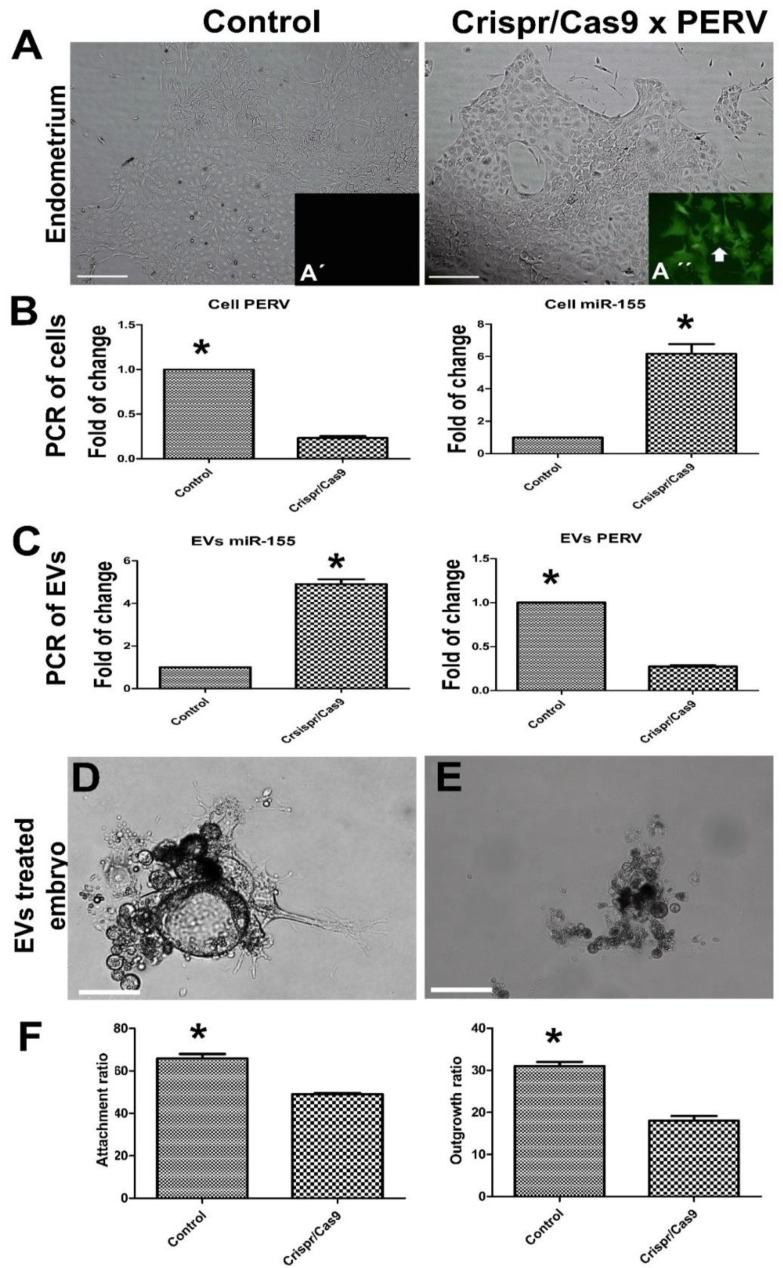
The impacts of targeting porcine endogenous retrovirus (PERV) expression in porcine endometrium through CRISPR/Cas9 on the derived EVs and the subsequent embryo development. (**A**) Endometrium was transfected with CRISP/Cas9 vector for 24 h;(**A**’,**A**’’) to compare the green fluorescence protein expression (white arrow, see Appendix A for the vector details) in control and mutated cells, respectively. The resulting cells were used to isolate Endo-EVs as mentioned previously. Scale bar = 50 µm. PERV targeted endometrium showed 5-fold and 4.5-fold reduction in the mRNA expression in both endometrium (**B**) and their derived EVs (**C**), respectively, while they showed upregulated miR-155 about 6-fold and 5-fold in endometrium (**B**) and their derived EVs (**C**), respectively. (**D**); (**E**) On day-5, embryo treated with PERV-targeted and PERV-diminished EVs showed low percentages of embryo attachment and outgrowths (**F**). Values were expressed as means ± S.E.M. and the difference between the two groups was compared with Student’s *t*-test. Values denoted by an asterisk (*) were considered statistically significant (*p* < 0.05).

**Table 1 cells-11-03178-t001:** Sequences for miR-155 mimic and miR-155 inhibitor [21].

Name	Sequence (5′ → 3′)
miRNA-155 inhibitor	UUAAUGCUAAUCGUGAUAGGGG
miRNA-155 mimic sense	UGGUGCAGGUUUAAUGCUAAUCGUGAUAGGGGUUUA
miRNA-155 mimic anti-sense	GUGCUGAUGAACACCUAUGCUGUUAGCAUUAAUCUUGCGCUA

**Table 2 cells-11-03178-t002:** Primers used for RT-PCR and RT-qPCR.

Name	Sequence 5′ → 3′		Product Size	Accession No.
	Forward	Reverse		
miR-100-p	AAACCCGTAGATCCGAACT	CAAGCTTGTGCGGACTAATA	43	NR_029515.1
miR-132-p	GTCTCCAGGGCAACCGTG	CGACCATGGCTGTAGACTGT	70	LM608489.1
miR-155	GCGGTTAATGCTAATCGTGATA	CGAGGAAGAAGACGGAAGAAT	65	LM608611.1
U6	GCTTCGGCAGCACATATACTAAAAT	CGCTTCACGAATTTGCGTGTCAT	89	NR_004394.1
Bax	GAGAGACACCTGAGCTGG	AGTTCATCTCCAATGCGC	165	XM_013998624.2
Bcl2	GTTGACTTTCTCTCCTACAAG	GGTACCTCAGTTCAAACTCAT	277	NM_214285.1
CDK2	GCTTCAGGGGCTAGCTTTTT	AGCCCAGAAGGATTTCAGGT	197	NM_001285465.1
CTNNB1	CCATTCCATTGTTTGTGCAG	GTTGCCACACCTTCATTCCT	175	NM_214367.1
GAPDH	ACACTCACTCTTCTACCTTTG	CAAATTCATTGTCGTACCAG	90	DQ845173.1
IFNG	CCATTCAAAGGAGCATGGAT	TTCAGTTTCCCAGAGCTACCA	76	NM_213948.1
PERV	TCCGTGCTTACGGGTTTTAC	TTTCTCCCAGAGCCTCCATA	388	XM_021074788.1
PTPN1	TCTCAAGAAACTCGAGAGAT	TCAGCCAGACAGAAGGTC	194	XM_021077277.1
Zeb1	ACGGATGCAGCAGATTGTGA	CCGGGTAACACTGTCTGGTC	71	XM_021064196.1
Zeb2	GACAATGTAGTGGACACGGGT	GGGGAGCACTCCTGGTT	131	XM_021076508.1
Universal stem-loop primer	GAAAGAAGGCGAGGAGCAGATCGAGGAAGAAGACGGAAGAATGTGCGTCTCGCCTTCTTTCNNNNNNNN			

U6: RNU6-1 RNA, U6 small nuclear 1 (house-keeping snRNA; https://www.ncbi.nlm.nih.gov/gene/26827, accessed on 16 August 2022); Bax: BCL2-associated X, apoptosis regulator (causes apoptosis; https://www.ncbi.nlm.nih.gov/gene/396633, accessed on 16 August 2022); Bcl2: BCL2 apoptosis regulator (antiapoptotic; https://www.ncbi.nlm.nih.gov/gene/100049703, accessed on 16 August 2022); CDK2: Cyclin-dependent kinase 2 (cell cycle regulation [38]); CTNNB1: β-catenin (cell attachment molecule and trophoblast invasion [39,40]); IFNG: interferon gamma (implantation signal produced by porcine embryo [41]); GAPDH: glyceraldehyde-3-phosphate dehydrogenase (house-keeping gene); PERV: porcine endogenous retrovirus; PTPN1: protein tyrosine phosphatase non-receptor type 1 (cell adhesion, migration, growth, differentiation, and mitotic cycle [42]); Zeb1: zinc finger E-box binding homeobox 1 (epithelial–mesenchymal transition and trophoblast cell differentiation [43]); Zeb2: zinc finger E-box binding homeobox 2 (epithelial–mesenchymal transition and trophoblast cell differentiation [44,45]).

**Table 3 cells-11-03178-t003:** The top 20 proteins identified in endometrium-derived extracellular vesicles.

UniProt Accession	Description	Mol %
A0A287B5W2	Trypsinogen isoform X1	6.0325
A0A287AEL2	Keratin 14	5.0814
F1SGG6	Keratin 5	4.831
A0A287A0Q6	Tyrosine 3-monooxygenase/tryptophan 5-monooxygenase activation Protein zeta	4.6558
A0A287ANZ8	Thy-1 membrane glycoprotein	4.1051
A0A286ZT13	Albumin	4.03
P20112	SPARC	3.5544
Q28944	Procathepsin	3.0288
A0A5G2QTF5	Thioredoxin	2.8536
A0A287AHS0	Calmodulin 3	2.8285
A0A287BA49	Keratin 5	2.7785
A0A287A8S8	Phosphopyruvate hydratase	2.7284
F1SGG3	Keratin, type II cytoskeletal 1	2.7034
I3LDS3	Keratin 10	2.6783
A0A287AHK1	Vitamin D-binding protein	2.6283
K7GQ95	S100 calcium binding protein A2	2.3529
A0A5G2QSE8	Keratin 3	2.3029
A0A287BHY5	Keratin 2	1.7772
A0A5G2QUE0	Antithrombin-III	1.7522
A0A287ATD0	Keratin 75	1.577

## Data Availability

The data supporting the findings of this study are included in the manuscript and the Appendix A.

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
