# Peer review of "ROCK Inhibitor (Y-27632) Abolishes the Negative Impacts of miR-155 in the Endometrium-Derived Extracellular Vesicles and Supports Embryo Attachment"

_cells, 2022, doi:10.3390/cells11193178_

Round 1

Reviewer 1 Report

As the first attempt to understand the roles of EVs cargo in determining embryo developmental competence and mediating molecular signaling between the embryo and the endometrium in porcine, this study is valuable.

This model will certainly help to improve and establish an ideal culture system to comprehend the early embryonic stem cell differentiation in porcine, specialy.

Some small questions and suggestions to improve what has been described:

Line 114: Why use fetal bovine serum in swine cell culture? Could it be replaced by another protein, synthetic or porcine source?

Lines 154 - 172: item 6 and its sub-items need spacing between paragraphs.

Line 229: Why use Nanodrop to quantify total RNA from so few embryos? What was the average quantification obtained in a pool of 5 embryos from the 4 replicates? What guarantee does this method or equipment give us for such small amounts of material?

Line 248: All the primers pairs had the same anneling condition? And about the efficient coeficient, which average was considered?

Line 541: Some complement is missing from the final sentence, or words were left by mistake. To correct.

The size ratio of symbols and letters (captions) in figures and graphs is not matching. Could reduce the font size, and standardize this size for everyone.

Why did you choose to use SEM and not SD, on statistics, as usual?

Author Response

As the first attempt to understand the roles of EVs cargo in determining embryo developmental competence and mediating molecular signaling between the embryo and the endometrium in porcine, this study is valuable.

This model will certainly help to improve and establish an ideal culture system to comprehend the early embryonic stem cell differentiation in porcine, specialy.

Response:

Response: We acknowledge the efforts, comments, and suggestions of the reviewer and we considered all these comments and suggestions when revising our manuscript.  

Some small questions and suggestions to improve what has been described:

Q1- Line 114: Why use fetal bovine serum in swine cell culture? Could it be replaced by another protein, synthetic or porcine source?

R1- We thank the reviewer for this notice. As we referred to in the methods, we followed the method of Zhang et al. (1991) and other researchers who used fetal bovine serum as a standard protein supplement in endometrial cell culture.

Ref: Zhang Z, Paria BC, Davis DL. Pig endometrial cells in primary culture: morphology, secretion of prostaglandins and pro-teins, and effects of pregnancy. Journal of Animal Science. 1991;69:3005-15.

Q2- Lines 154 - 172: item 6 and its sub-items need spacing between paragraphs.

R2- We thank the reviewer for this correction. We edited the manuscript accordingly.

Q3- Line 229: Why use Nanodrop to quantify total RNA from so few embryos? What was the average quantification obtained in a pool of 5 embryos from the 4 replicates? What guarantee does this method or equipment give us for such small amounts of material?

R3- We thank the reviewer for this technical note. We run this method in many of our papers, and the average total RNA concentration ranges between 10-20 ng/microL. We used the kit which enabled us to isolate the minute amounts of RNA (RNeasy Micro Kit (Qiagen GmbH, Hilden, Germany). Quantity and quality of RNA were evaluated by a NanoDropTM spectrophotometer (Thermo fisher) with OD ratios 260/280 and 260/230 of values > 1.8. We also followed the method of Varkonyi-Gasic et al. (2007) who used it for the small quantity or single cells extracted RNA to increase the efficiency and quality of the transcribed cDNA. Pulsed reverse transcription was conducted to generate complementary DNA (cDNA). Twenty nanograms of total RNA in a 20-μl reaction volume were used as a template for cDNA synthesis in 60 cycles of 2 min at 16 °C, 1 min at 37 °C, and 0.1 s at 50 °C, and a final inactivation at 85 °C for 5 min.

Ref: Varkonyi-Gasic, E., Wu, R., Wood, M. et al. Protocol: a highly sensitive RT-PCR method for detection and quantification of microRNAs. Plant Methods 3, 12 (2007). https://doi.org/10.1186/1746-4811-3-12

Q4- Line 248: All the primers pairs had the same anneling condition? And about the efficient coeficient, which average was considered?

R4- We thank the reviewer for this technical note. We designed our primers following Primer 3 website and all primers were tested through a gradient real-time PCR before the experiments to have a single peak in the melting curve and the optimized annealing temperature. Please find attached some pictures of the melting curve. We referred to this in the revised manuscript. We hope we clarified the point to you.

Q5- Line 541: Some complement is missing from the final sentence, or words were left by mistake. To correct.

R5- We apologize for this typo. We deleted it.

Q6- The size ratio of symbols and letters (captions) in figures and graphs is not matching. Could reduce the font size, and standardize this size for everyone.

R6- Thank you for the great suggestion. We adjusted the font size and labels accordingly.

Q7- Why did you choose to use SEM and not SD, on statistics, as usual?

R7- We are very sorry for this typo. In fact, the default setting of the software GraphPrism was adjusted at the means +/- SD in the case of the ANOVA test and +/- SEM in the t-test. We corrected this throughout the manuscript.

------------------------

We hope our response and the edited manuscript meet the quality for publication.

Thank you.

Reviewer 2 Report

The study by Saadeldin et al.  used porcine parthenogenic embryos to study the effect of endometrium-derived extracellular vesicles (EVs) and their cargos on embryo attachment. Specifically, they find that one of the EV cargos, miR-155, hampers embryo attachment and this negative effect can be alleviated by ROCK inhibitor (Ri). While their results are potentially interesting to understand maternal-embryo crosstalk, several aspects (detailed below) need to be addressed.

1.     There are studies about the role of EVs in embryo implantation in different species. The authors should mention these studies in their introduction so that the readers can have a better understanding of the research background. Also, they should explain their rational of using Ri.

2.     The overall quality of their parthenogenic embryos is not optimal when comparing with that reported by other groups. For example, the cell number (per blastocyst) in their control group (Figure 4c) is only around 20 while many other groups can get around 50. Could their suboptimal culture system affect any downstream experiments?

3.     Figure 2, what is the negative control? Why do they use only one negative control for all the PCR reactions? They should use a negative control (blank template) for each gene. It is difficult to tell if the gel band for Caternin in Endometrium is real amplification or if it is just primer dimers because a similar band can also be observed in the negative control.

4.     Line 360-370. It will be easier for the readers to understand if the authors can explain the experimental design and different treatment groups before talking about the results.

5.     Line 362-363. It should be “…apoptosis (2.37% vs 26.1%, p<0.01)…”

6.     Figure 4C. Ri treatment alone can inhibit embryo development?

7.     Figure 4D. Cell apoptosis in the control group is very high (26.1%). Again, is it due to their suboptimal embryo culture system (point 2)?

8.     From Figure 4B, 4C and 4D, it seems that Ri alone does not affect embryo attachment (4B) and hampers embryo development (4C). Although RI alone can decrease cell apoptosis (4D), if comparing between EVs and EVs+RI, it seems that EVs alone can already result in a large reduction in cell apoptosis. Taken all three results together, what is the logic behind their choice of RI for subsequent experiments?

9.     Figure 5A, the images are not clear. Probably they can do simple DAPI staining to show the results better. Since the days labelled in Figure 5A are extra days after day 7, they should label Day 9, Day 10 and Day 12 respectively to avoid confusion.

10.  Figure 6, with so few cells/embryo in both groups, it is difficult to tell if these are normal embryos. For the staining (especially for OCT4), it is difficult to tell if the staining is positive since there is not so much overlap between DAPI and antibody staining.

11.  Figure 7. It will be better for the readers to understand if they can explain in the text why they chose to examine these genes and relate the gene expression results to the effect of EVs+RI on embryo development (Figure 4 and 5). This should also be discussed in the discussion.

12.  Figure 8 B, C, D are too small. Adjust the dimensions of the plots.

13.  The Y-axis label of Figure 8B should be “attachment ratio” not “attachment %”.

14.  Figure 8C, the control EVs cell number is only 12. But in Figure 4C, they show the cell number in EVs groups is 26. How reproducible are their results?

15.  Figure 9B and C. Why do RI+MiR-155 mimic have better attachment and larger cell number than RI alone since MiR-155 alone can inhibit both aspects? As mentioned in point 7, Ri alone does not affect embryo attachment (4B) and decreases cell number. But in Figure 9B and C, why it improves both aspects after combining with MiR-155 mimic?

16.  miR-155 and PERV seem to be mutually repressive (Figure 10B). They should check the expression of PERV after miR-155 inhibition or miR-155 mimic treatment.

17.  Figure 10D. What is the difference between the two images?

18.  Discussion. They should discuss point 2, 7, 8, 11, 14 and 15 in their discussion.

19.  Line 578-584. This is perspective not conclusion.

Author Response

Reviewer 2:

The study by Saadeldin et al.  used porcine parthenogenic embryos to study the effect of endometrium-derived extracellular vesicles (EVs) and their cargos on embryo attachment. Specifically, they find that one of the EV cargos, miR-155, hampers embryo attachment and this negative effect can be alleviated by ROCK inhibitor (Ri). While their results are potentially interesting to understand maternal-embryo crosstalk, several aspects (detailed below) need to be addressed.

Response: We acknowledge the efforts, comments, and suggestions of the reviewer and we considered all these comments and suggestions when revising our manuscript. The manuscript has been greatly improved after addressing all these comments and suggestions.

Q1.     There are studies about the role of EVs in embryo implantation in different species. The authors should mention these studies in their introduction so that the readers can have a better understanding of the research background. Also, they should explain their rational of using Ri.

R1. We appreciate the reviewer’s suggestion. We added some details in the introduction accordingly.

Q2.     The overall quality of their parthenogenic embryos is not optimal when comparing with that reported by other groups. For example, the cell number (per blastocyst) in their control group (Figure 4c) is only around 20 while many other groups can get around 50. Could their suboptimal culture system affect any downstream experiments?

R2. We thank the reviewer for highlighting this important point. We are working with parthenogenetic activation (PA) of porcine oocytes since 2004 and we are completely aware of the cell numbers of parthenogenetic embryos. Our cell number on day 7 coincides with many other research groups such as Randall S. Prather’s group who showed the average cell number of PA embryos is about 29 cells and tended to be reduced with the prolongation of culture to day 8.

I quoted the results of Dr. Prather’s group (Hao, Y., Lai, L., Mao, J., Im, G. S., Bonk, A., & Prather, R. S. (2004). Apoptosis in parthenogenetic preimplantation porcine embryos. Biology of reproduction70(6), 1644-1649) for your consideration (Attached file).

In addition, they noted that apoptosis reached about 29% by day-8 of culture (Attached file). That answer can clarify the questions that were raised by the reviewer in the subsequent questions (Q7, Q8, and Q14).

Q3.     Figure 2, what is the negative control? Why do they use only one negative control for all the PCR reactions? They should use a negative control (blank template) for each gene. It is difficult to tell if the gel band for Caternin in Endometrium is real amplification or if it is just primer dimers because a similar band can also be observed in the negative control.

R3. We thank the reviewer for this insightful note. We used negative control (cDNA template free reaction) for each gene in our initial assessment of the primers. We just provided a representative of the negative cDNA reaction. Regarding catenin expression, we sometimes find the band as faint or absent through our cycling and amplification conditions (size is 175 bp). In addition, we excluded the presence of dimers through the real-time PCR reaction which showed a single peak in the melting curve. We cropped the bands to avoid confusion for the readers. Therefore, we wrote in the illustration of the figure (-/+) because we can detect its expression in some replicates. We here provide an image of gel electrophoresis from the different replicates of the PCR.

We also provide the melting curves of all primers used in the study (Attached file).

Q4.     Line 360-370. It will be easier for the readers to understand if the authors can explain the experimental design and different treatment groups before talking about the results.

R4. Thank you for the useful suggestion. We added this sentence accordingly.

Q5.     Line 362-363. It should be “…apoptosis (2.37% vs 26.1%, p<0.01)…”

R5. Thank you so much for the correction. We edited it accordingly.

Q6.     Figure 4C. Ri treatment alone can inhibit embryo development?

R6. We thank the reviewer for this important note. We wrote a detailed review about the effects of ROCK and its inhibition on the different stages of embryonic development and most of the studies showed the negative impacts of rock inhibition on embryonic development. However, ROCK inhibition is critical for embryonic stem cell fate determination and differentiation. Also, it depends on the species as we showed in the review (Saadeldin et al., 2021) “Rocking the boat: the decisive roles of Rho kinases during oocyte, blastocyst, and stem cell development. Frontiers in Cell and Developmental Biology8, p.616762)” and supported by the findings of Motomura and Hou (Motomura et al. 2017. A Rho-associated coiled-coil containing kinases (ROCK) inhibitor, Y-27632, enhances adhesion, viability and differentiation of human term placenta-derived trophoblasts in vitro. Plos One 12(5):e0177994 & Hou et al. 2015. The Efficient Derivation of Trophoblast Cells from Porcine In Vitro Fertilized and Parthenogenetic Blastocysts and Culture with ROCK Inhibitor Y-27632. Plos One 10(11):e0142442. This will also answer Q8 and some other questions.

Q7.     Figure 4D. Cell apoptosis in the control group is very high (26.1%). Again, is it due to their suboptimal embryo culture system (point 2)?

R7. I think we clarified this point in Response to Q2.

Q8.     From Figure 4B, 4C and 4D, it seems that Ri alone does not affect embryo attachment (4B) and hampers embryo development (4C). Although RI alone can decrease cell apoptosis (4D), if comparing between EVs and EVs+RI, it seems that EVs alone can already result in a large reduction in cell apoptosis. Taken all three results together, what is the logic behind their choice of RI for subsequent experiments?

R8. I think we clarified this point in Response to Q6. Moreover, according to our results and the previous results of (Hou et al., 2015; Motomura et al., 2017) as shown in response to Q6, we propose a synergistic effect of RI to enhance the action of Endo-EVs through antagonizing the actions of their contents of miR-155.

Q9.     Figure 5A, the images are not clear. Probably they can do simple DAPI staining to show the results better. Since the days labelled in Figure 5A are extra days after day 7, they should label Day 9, Day 10 and Day 12 respectively to avoid confusion.

R9. We thank the reviewer for this suggestion. In this experiment, we followed the embryonic outgrowths daily for 5 days, so we did not use DAPI staining to not alter the DNA of the cells. According to your suggestion, we changed the labeling accordingly.

Q10.  Figure 6, with so few cells/embryo in both groups, it is difficult to tell if these are normal embryos. For the staining (especially for OCT4), it is difficult to tell if the staining is positive since there is not so much overlap between DAPI and antibody staining.

R10. We are sorry for giving a poor-quality image of the Endo-EVs+RI cells. We replaced it with a good quality one.

Q11.  Figure 7. It will be better for the readers to understand if they can explain in the text why they chose to examine these genes and relate the gene expression results to the effect of EVs+RI on embryo development (Figure 4 and 5). This should also be discussed in the discussion.

R11. We thank the reviewer for this suggestion. We showed the functions of these genes in the table with the corresponding references and also clarified this in the discussion and the computational analysis in L569-582.

We also performed GeneMania bioinformatic gene-to-gene interaction analysis including the different isoforms of ROCK (ROCK1 and ROCK2). Results indicate an existing physical interaction, shared protein domains, common pathways, and co-expression of the studied genes (Attached file).

Q12.  Figure 8 B, C, D are too small. Adjust the dimensions of the plots.

R12. We replaced the figure accordingly.

Q13.  The Y-axis label of Figure 8B should be “attachment ratio” not “attachment %”.

R13. R12. We replaced the figure accordingly and with other figures with the same Y-axis label.

Q14.  Figure 8C, the control EVs cell number is only 12. But in Figure 4C, they show the cell number in EVs groups is 26. How reproducible are their results?

R14. We are thankful for this notion. As we mentioned above, time plays an important role in affecting the porcine embryonic cell number in prolonged culture. In Figure 4, cells were counted 36 h after the culture, while in Fig. 8 the cells were counted after 5 days of transfection and culture. These two factors can affect the cell count. So, there are no contracted results in the presented work.

Moreover, in the current work, we used feeder-free culture conditions and drop in cell number is expected as we have shown earlier in 2015 (Saadeldin et al. Dev Growth Differ. 2015 Jun;57(5):362-368. doi: 10.1111/dgd.12215). We think the current results provide a step forward to optimizing the embryonic stem cell culture of porcine in feeder-free culture conditions.

Q15.  Figure 9B and C. Why do RI+MiR-155 mimic have better attachment and larger cell number than RI alone since MiR-155 alone can inhibit both aspects? As mentioned in point 7, Ri alone does not affect embryo attachment (4B) and decreases cell number. But in Figure 9B and C, why it improves both aspects after combining with MiR-155 mimic?

R15. As we clarified above, RI antagonized the effects on miR-155-induced apoptosis, attachment, and cell numbers. We propose a synergistic effect of RI to enhance the action of Endo-EVs through antagonizing the actions of their contents of miR-155.

Q16.  miR-155 and PERV seem to be mutually repressive (Figure 10B). They should check the expression of PERV after miR-155 inhibition or miR-155 mimic treatment.

R16. We thank the reviewer for highlighting this point. In our preliminary trials, we couldn’t observe any change in PERV mRNA expression after using a miR-155 mimic or inhibitor. We can expect that miRNA-155 is not targeting the 5′ UTR, coding sequence, and gene promoters of the PERV and doesn’t necessarily affect its expression from the gene. However, further experiments are needed to prove this regard.

Q17.  Figure 10D. What is the difference between the two images?

R17. We labeled the two figures and replaced the control group with a suitable representative image.

Q18.  Discussion. They should discuss point 2, 7, 8, 11, 14 and 15 in their discussion.

R18. We edited the discussion to clarify this part.

Q19.  Line 578-584. This is perspective not conclusion.

R19. We thank the reviewer for this comment. We added a conclusion statement.

------------------------

We hope our response and the edited manuscript meet the quality for publication.

Thank you.

REFERENCES

 Hou, D., Su, M., Li, X., Li, Z., Yun, T., Zhao, Y., Zhang, M., Zhao, L., Li, R., Yu, H., and Li, X. (2015). The Efficient Derivation of Trophoblast Cells from Porcine In Vitro Fertilized and Parthenogenetic Blastocysts and Culture with ROCK Inhibitor Y-27632. Plos One 10. 10.1371/journal.pone.0142442.

Motomura, K., Okada, N., Morita, H., Hara, M., Tamari, M., Orimo, K., Matsuda, G., Imadome, K.-I., Matsuda, A., Nagamatsu, T., et al. (2017). A Rho-associated coiled-coil containing kinases (ROCK) inhibitor, Y-27632, enhances adhesion, viability and differentiation of human term placenta-derived trophoblasts in vitro. Plos One 12. 10.1371/journal.pone.0177994.

Saadeldin, I.M., Tukur, H.A., Aljumaah, R.S., and Sindi, R.A. (2021). Rocking the Boat: The Decisive Roles of Rho Kinases During Oocyte, Blastocyst, and Stem Cell Development. Frontiers in Cell and Developmental Biology 8. 10.3389/fcell.2020.616762.

Round 2

Reviewer 2 Report

The revised manuscript has been improved. Below are a few minor points:

1.     Line 244, the 260/280 ratio for RNA should be >= 2.0, since the authors use 1.8 as the threshold, can they rule out the possibility of DNA contamination?

2.     Line 382, remove “wither”.

3.     Figure 8D, how was the apoptosis ratio calculated, by calculating signal intensity or by counting positive cells from the TUNEL assay? How many embryos were used for the calculation? If by counting cell number, considering the total cell number per embryo in both groups, the results in Figure 8D mean there is basically no apoptosis in most embryos. Is this the case? Is this also the reason why the authors used the negative-staining images for both groups in Figure 8A (TUNEL)?

Author Response

The revised manuscript has been improved. Below are a few minor points:

We thank the reviewer for their thoughtful suggestions. 

1- Line 244, the 260/280 ratio for RNA should be >= 2.0, since the authors use 1.8 as the threshold, can they rule out the possibility of DNA contamination?

R1. We thank the reviewer for this comment. We did not measure the DNA in the samples, however, following the kit manufacturer’s instructions we used DNase to exclude any DNA contamination. We edited the text and provided the Kit Catalo No. for the clarification.

2- Line 382, remove “wither”.

R2. We are sorry for this typo. We deleted it.

3- Figure 8D, how was the apoptosis ratio calculated, by calculating signal intensity or by counting positive cells from the TUNEL assay? How many embryos were used for the calculation? If by counting cell number, considering the total cell number per embryo in both groups, the results in Figure 8D mean there is basically no apoptosis in most embryos. Is this the case? Is this also the reason why the authors used the negative-staining images for both groups in Figure 8A (TUNEL)?

R3. We agree with the reviewer, we counted the positive green fluorescent or apoptotic cells and calculated the percentage of apoptotic cell to the total cell numbers for each replicate. We clarified this in the edited text. As you can see, the apoptosis % in Fig 8 is around 3-1% and the total number of the cells ranges from 12-31 so, the representative image doesn’t show signals to avoid the readers confusion. However, as we showed in materials and methods (13. Statistical analysis) we used 20 embryos and 3 replicates for doing this analysis.